# Green Extractants in Assisting Recovery of REEs: A Case Study

**DOI:** 10.3390/molecules28030965

**Published:** 2023-01-18

**Authors:** Dorota Kołodyńska, Katarzyna Burdzy, Steffi Hunger, Andreas Aurich, Yongming Ju

**Affiliations:** 1Department of Inorganic Chemistry, Institute of Chemical Sciences, Faculty of Chemistry, Maria Curie-Skłodowska University, Maria Curie-Skłodowska Sq. 2, 20-031 Lublin, Poland; 2Department Centre for Environmental Biotechnology (UBZ), Helmholtz-Centre for Environmental Research-UFZ, Permoserstrasse 15, 04318 Leipzig, Germany; 3Nanjing Institute of Environmental Sciences, Ministry of Ecology and Environment (MEE), Nanjing 210042, China; 4The Key Laboratory of Water and Air Pollution Control of Guangdong Province, South China Subcenter of State Environmental Dioxin Monitoring Center, South China Institute of Environmental Sciences, Ministry of Ecology and Environment (MEE), Guangzhou 510655, China

**Keywords:** rare earth, heavy metal ions, leaching, citric acid, tartaric acid, ethylenediaminedisuccinic acid

## Abstract

The recycling of REEs from the end of life (EoL) products, such as nickel metal hydride batteries (NiMH), offers great opportunities for their supply in Europe. In the presented paper, the application of ‘green’ extractants such as citric (CA), metatartaric (TA), and ethylenediaminedisuccinic acid (EDDS) (also with H_2_O_2_ addition) for the recovery of REEs was studied. The studies were conducted considering the effects of the phase contact time, the initial concentration of CA, TA, and EDDS, as well as H_2_O_2_, pH, and temperature. It was found that the addition of TA to the CA solution meant that higher rates of metal ion binding and, thus, leaching was observed. The optimal conditions were obtained in the system: CA-TA and H_2_O_2_ for the concentration 0.6M-0.3 M-2%.

## 1. Introduction

Scandium, yttrium, lanthanum, and lanthanides are called rare earth elements and are often referred to as REE or Ln^3+^. They belong to group three of the periodic table of elements. Yttrium and lanthanum are similar to lanthanides and occur together in nature. The major yttrium and lanthanum ores are monazite (Ce, La, Nd, Pr, Th, Y)PO_4_ and bastnasite (Ce, La, Y)CO_3_F; however, a ‘yttrium-rich’ mineral might contain ≤1% Y(III) and ‘lanthanum-rich’ could contain up to 35% La(III). The elements from La(III) to Lu(III) are characterized by the 3+ oxidation state, and the chemistry is mostly that of the Ln^3+^ ion [1]. They have extremely similar physicochemical properties and behaviours in the environment, so they constitute a relatively cohesive group. The ‘lanthanide contraction’ is also typical (decrease in the atomic and ionic radii) of the gadolinium break. The ionic radius of REE ions drops from 1.061 Å for La(III) to 0.848 Å for Lu(III). REEs are characterized by low electronegativity and the increasing stabilisation of the 4f, 5d, and 6s orbitals. As for the coordination number, most of them form complexes within 6–12. REEs are used in various types of applications and increase gradually, which means that their consumption increases and reserves decrease [2]. In addition to the annual growth rate of global REE needs, an extremely large demand is expected for Nd and Dy, up to 700% and 2600%, respectively, in the next 25 years [3]. The other aspect is the fact that some mineral resources are inaccessible to mining because of environmental restrictions and government policies.

The recovery and recycling of REE from wastes generally consumes less energy and results in a smaller environmental impact compared with their primary production from the ores. It was estimated that 1 ton of rare earth generates approximately 8.5 kg of fluorine, 13 kg of dust, 60,000 m^3^ of waste gas containing hydrofluoric acid, and 200 m^3^ of acidic sewage. Therefore, recycling is an important factor in the availability of REE. This is also considered a valuable way of balancing the REE market and, thus, a positive and effective technique to reduce the harmful environmental effects of REE mining. 

What is more important is that REEs of high-purity play a significant role in many areas of contemporary techniques. They are used in lasers, magnets, phosphors, motion picture projectors, X-ray intensifying screens, wind turbines, HEVs and all-electric vehicles, light-emitting diodes, and fluorescent lamps. The addition of the pyrophoric mixed rare-earth alloy called Mischmetal or lanthanide silicates improves the strength and workability of small steel alloys. They also have many scientific applications. Therefore, the preparation of high purity rare earth elements is crucial for such technologies, especially so they are present altogether and with heavy metal ions [4].

More than ten years ago, the supply of REE through quotas, licenses, and taxes was limited by China. In 2010, China cut its production quota for REE concentrates by 25% and its export by 37%. At that time, the Chinese share of supply increased from 21% in 1985 to 97% in 2005–2011. In 2018, this share decreased to 71% (excluding undocumented data). The concerns about the reduction in REE availability outside of China have intensified with the increase in the prices of REE products for export by up to 600% in 2011 [5]. Consequently, the other countries began to look for REE deposits and to promote new ideas to protect, recycle, and find substitutes for these elements based on the circular economy rules (Figure 1) [6,7,8]. For example, since the 1980s, (NH_4_)_2_SO_4_ has been used to replace NaCl as the solvent to obtain high-grade rare earth concentrates (REO 92%). In the early 1990s, it was replaced by the cheap leaching procedure (the so-called solution mining). 

In addition to the primary production, a range of secondary REEs production processes containing: (i) the recovery of REEs from the phosphoric acid industry, (ii) fluorescent lamps, (iii) the recycling of REEs magnet, as well as (iv) e-wastes was established. This approach is consistent with Goal 12 of the United Nations 2030 Agenda for Sustainable Development Goals. The European Raw Materials Alliance (ERMA), as part of the Action Plan on Critical Raw Materials, was announced on 3 September 2020, based on three pillars: (i) ensuring a level playing field in the access to the resources of the world third countries, (ii) fostering a sustainable supply of raw materials from the European sources, and (iii) boosting resource efficiency and promoting recycling. 

The recycling of REEs from wastes is an important step towards building a sustainable REE supply chain, especially when you consider the fact that the current annual world production of REE oxides is about 126 kt. Globally, only 1% of REEs are recovered and recycled from EoL products [8]. 

According to Habashi [9], the main resource of phosphorous in the phosphoric acid industry is based on apatite Ca_5_(PO_4_)_3_(Cl, F OH) with 0.1–1% REE (due to the substitution of Ca(II) and being balanced by Na(I) or as the monazite inclusion). This uses 250 million phosphate rocks with an average REEs content of 0.046 wt%. The other one is phosphogypsum: a waste product of phosphoric acid production. In the dihydrate process, 70–80% of the REEs from the phosphate rock migrates to the phosphogypsum waste stream, but in hemihydrate, the hemi-dihydrate processes to more than 95%. 

As for the linear (LFLs) and compact fluorescent lamps (CFLs), the REO content (g/unit) of various energy-efficient light types depends on the phosphor used for their production. They contain Y, La, Ce, Tb, and Eu in the form of red (Y_2_O_3_:Eu, YOX), blue (BaMgAl_10_O_17_:Eu^2+^, BAM), and green (CeMgAl_11_O_19_:Tb^3+^, CTMA/CAT or LaPO_4_:Ce^3+^, Tb^3+^, LAP) phosphors with REEs 10–20% content. Chloroapatite (Sr, Ca, Ba, Mg)_5_(PO_4_)_3_Cl: Eu^2+^ is a less common blue phosphor. Many fluorescent lamps and those made by some manufacturers outside of China contain halophosphates (Sr, Ca)_10_(PO_4_)_6_(Cl, F)_2_:Sb^3+^, Mn^2+^ or its mixture with Y_2_O_3_:Eu^3+^. They are recycled by removing the glass, phosphor decomposition during alkaline fusion at 800 °C, and washing excess alkali and sodium aluminate followed by REEs leaching [10]. For the REEs recovery, HNO_3_, HCl, or H_2_SO_4_ with the Y, Eu extraction over 90% at the 1−4 M acid concentration, a 1−24 h leaching time, and a temperature of 50−110 °C are mainly used. Two-step processing, i.e., the Eu-Y leaching followed by the microwave treatment, was found to be better compared to the single-step process. According to Shukla and Dhawan [11], the addition of NaOH and H_2_SO_4_ enhanced the method significantly. The acid leaching of the phosphor feed yielded 90% Y and 84% Eu extraction with minimal Ce, Tb, and La amounts. The microwave exposure facilitated the release process with 65% La, 3% Ce, 35% Eu, and 50% Tb extraction. Moreover, Na_2_CO_3_ can be also used for REEs recovery from green phosphors. A pioneer in this field is the Japan Oil, Gas, and Metals National Corporation.

Another source of RREs is various types of optical glass, with a total world production of 20,000 tonnes per year for the production of camera lenses, microscopes, binoculars, or microscopes with a high refractive index and low dispersion. It contains more than 40% by weight of La_2_O_3_ with Y_2_O_3_ and Gd_2_O_3_. The recycling of worn-out optical glasses can contribute to the recovery of about 1.6 thousand tonnes of REO per year.

The high concentration of rare earth (mainly in the form of oxides) in the spent glass polishing material enables an easier and resource-saving production predominantly of CeO_2_ and La_2_O_3._ Additionally, Fe_2_O_3_, ZrO_2_, TiO_2_, Al_2_O_3_, etc. CeO_2_ or polishing agents with CeO_2_ is the most widely used, and today they are commercially traded with the brands of Auerpol^®®^, Cerox^®®^, Opaline^®®^, Lensmax^TM^, Super Ce-Rite^®®^, and Shorox^TM^ [12]. The REEs recovery from glass polishing is made using oxalic or citric acid leaching (pH–1.5). However, to remove silica, flotation aided by sonication is proposed in the alkali leaching (pH–11.5) with roasting at 600 °C for 2 h. REEs can also be removed using sulfuric acid (3 mol/L), S/L 1:10, 60 °C by 3 h [13]. REE sulfates’ solubility decreases with increasing temperature due to the exothermic nature of their dissolution. The powder can be treated by 2 mol/L sulfuric acids at 90 °C for leaching La, Nd, Pr, and Ca and in the second stage with 12 mol/L H_2_SO_4_ at 120 °C. After leaching, Ce(SO_4_)_2_ is dissolved [14].

In the case of the NdFeB magnet scrap, multistep hydrometallurgical techniques and liquid metal extraction with high-energy consumption are usually applied. The most popular REE magnets applied in hard drives, cell phones, loudspeakers, air conditioning units, or cars are based on the neodymium iron boron alloys Nd_2_Fe_14_B (with large amounts of Nd and small amounts of Pr, Gd, Tb, Dy) increase the temperature stability before demagnetization as well as for the other elements Co, V, Ti, Zr Nb, and Mo. Globally, about 600 mln hard drives are produced using about 6–12 thousand tonnes of NdFeB alloys. The recycling of REEs involves their dissolution in acids before the REE is precipitated out of the solution. The manufacture of NdFeB magnets also uses a series of grinding, cutting, and polishing operations that can result in the loss of REEs.

Coal fly ash (CFA) is a readily available waste product with strong environmental incentives and an established market for its beneficial reuse. It does not require extensive excavation. Most REEs in CFA are associated with the aluminosilicate glassy phase [15]. After the coal combustion, the components of the CFA are almost entirely inorganic, which can enrich the content of REEs and promote leaching efficiency. Rozelle et al. [16] showed that the extraction of REEs from the coal by-products is technically feasible, with a recovery rate of 89% using the eutectic solvent. After mixing with the appropriate amount of CFA with sodium carbonate at the 1:1 mass ratio of 860 °C for 30 min, the leaching levels of REE increase gradually with time and Ce exhibits the largest content. The leaching rate through the alkali fusion-leaching is equal to 58% [17]. Additionally, in [18], it was found that the fraction of critical REEs (Nd, Eu, Tb, Dy, Y, and Er) can be 34–38% of the total and is considerably higher than in the conventional ores (typically less than 15%).

Rare earth elements are widely used in various types of catalysts. For example, the catalytic cracking catalysts used in the petrochemical industry are another source of REEs with 3.5% by weight of REO, mainly La(III) and small amounts of Ce(IV) and Nd(III). Therefore, HNO_3_ and *Gluconobacter oxydans* can be used for their removal [19,20]. 

A possible way of maintaining continuity in the access to REEs is the recycling of nickel metal hydride batteries (NiMH). They are characterized by an energy density close to lithium-ion batteries (LIBs) and are two to three times greater than that of nickel-cadmium batteries (NiCd). In this type of battery, the electrochemical charge/discharge reactions proceed between the positive electrode (cathode, Ni(OOH) + H_2_O + e^−^ ⇄ Ni(OH)_2_ + OH^−^) and the negative electrode (anode, MH + OH^−^⇄ M + H_2_O + e^−^) which is composed of the hydrogen-absorbing alloy based on Mischmetal (M) e.g., MNi_3,55_Co_0,75_Mn_0,4_Al_0,3_. The electrodes are separated by a permeable membrane. As for REEs used for their production [21], mainly Ce (50–55%), La (18–28%), and Nd (12–18%) can be distinguished as La_0,62_Ce_0,27_Pr_0,03_Nd_0,08_. Therefore, at EoL, the NiMH batteries can act as a source of Ni, La, Ce, Nd, Pr, Zn, Mn, and Co. Leaching plays a key role in the recycling process. Due to the mislabelling of batteries, the separation of Cd(II) can also be problematic. What is more, the removal of Cd(II) is a crucial aspect during solvent extraction and Ni(II) electrowinning. Moreover, the current physicochemical methods for recycling the spent NiMH batteries are complex, energy-consuming, costly, and inefficient in the simultaneous recovery of base metals and REEs [22,23]. According to Agarwar et al. [24], only 1% of REEs are recovered and recycled from the EoL products, although, in the literature, there are several examples of recycling REEs from the NiMH batteries developed by Umicore and Rhodia. The condition for leaching depends on the used acid: HCl, HNO_3_, and H_2_SO_4_. These leaching agents release toxic gases, or their use is associated with the production of a large amount of wastewater. Other studies have proposed the use of bioleaching or ‘green leaching agents’ as an alternative to conventional hydrometallurgy [25]. Therefore, it is quite common to use more friendly reagents such as citric acid C_6_H_8_O_7_, malic acid C_4_H_5_O_6_, oxalic acid H_2_C_2_O_4_, ascorbic acid C_6_H_8_O_6_, aspartic acid C_4_H_7_NO_4_, succinic acid C_4_H_6_O_4_, and glycine C_2_H_5_NO_2_. In the paper by Rasoulnia [19], the leaching of rare earth elements and base metals from the spent NiMH batteries was conducted using D-gluconic acid sodium salt, 2-keto-D-gluconic acid hemicalcium salt, and 5-keto-D-gluconic acid potassium salt. The leaching experiments were performed at 27 °C for 14 days. A greater REEs leaching efficiency was obtained with gluconate, while 5-ketogluconate enabled more efficient base metal leaching for 100% Mn, 90.3% Fe, 89.5% Co, 58.5% Ni, 24% Cu, 29.3% Zn, and 56.1% total REEs [26]. REE fluorocarbonates are extracted using 50% K_2_CO_3_ as the leaching solution at 100 °C at the S/L (mL/g) ratio of 25; about 10% of the total REEs content of the considered sample is extracted within 1 h. On a laboratory scale, such alkaline leaching of REEs allows also for the recovery of K_2_CO_3_ from concentrated KOH in accordance with a circular flow [1]. 

As follows from the literature data, the recovery of REEs from magnet scrap, nickel metal hydride batteries, phosphors, glass polishing powders, catalytic cracking catalysts, and optical glass is cost-effective. The next challenge reported by Binnemans et al. [27] is a recovery rate of 75–100% in the laboratory scale experiments. However, there are no large-scale industrial rare earth recycling technologies. Moreover, after leaching, the REEs recovery can be accomplished by precipitation, solvent extraction, or ion exchange. 

Therefore, in the paper, research on the applicability of different types of complexing agents for the leaching of REEs from spent NiMH batteries is carried out. The main aim is determining the influence of the type of leaching system containing citric (CA), metatartaric (TA), ethylenediaminedisuccinic acid (EDDS) (also with the H_2_O_2_ addition) on the effectiveness of REEs leaching. The non-sterile citric acid was produced by as as described previously [28,29,30]. The studies were conducted considering the effects of the phase contact time, the initial concentration of CA, TA, and EDDS, as well as H_2_O_2_, pH, and temperature. After the comparisons, the most suitable treatment way was selected. To achieve the essential aim, semi-technical scale experiments are being planned. 

## 2. Results and Discussion

### 2.1. Physicochemical Characterization of Materials

The qualitative and semi-quantitative chemical analyses of several metallic components of the black battery mass have made it possible to estimate the X-ray fluorescence (XRF) method (Table 1). It was found that it contains, for example, 11.29% of iron and 30.77% of nickel as non-REE components and 4.31% of lanthanum, 1.84% of cerium, 0.13% of praseodymium, and 0.55%of neodymium per kg of the black battery mass. All elements were mainly determined as oxides at the level of wt.%. 

### 2.2. Leaching Parameters

It should be stressed that the choice of the leaching conditions depends on several factors, including the complexity of the system (type of REE ions, their oxidation state, type of complexing/leaching agent, etc.), pH, temperature range, and the overall economy of the system. Table 2 presents the leaching agents, time, temperature, and pH conditions used for the leaching of REEs from spent NiMH batteries.

In preliminary studies, it was found that different levels of leaching effectiveness depended on the time and type of the agent used—CA and CA with TA or EDDS (mixture (v/v) with or without the addition of H_2_O_2_) as well as temperature were obtained. The optimal parameters are collected in Table 2.

The pH of PLS obtained from the black battery mass using CA was equal to 4.45–4.47, and no precipitation yield was observed, which is consistent with the value previously reported in the literature. At this pH, the obtained results are presented in Figure 2 and Figure 3.

The first leaching agent was citric acid (CA) (Figure 4a,b). Similar to citric acid, citrates are used in the production of soft drinks and in the food industry as nutrients and food additives, acidity regulators, antioxidants, buffering, firming, preservatives, and stabilizing agents [31]. The structure of CA is presented in Figure 4.

The heavy metal complexes of citric acid are well described in the literature. In [32], it was found that three protonation constants of CA pK_1_, pK_2,_ and pK_3_ increased with the increasing temperatures 288 K, 298 K, and 308 K from 2.81, 2.91, 2.98; 4.14, 4.25, and 4.33 and 5.52, 5.65, and 5.74, respectively. Therefore, the corresponding enthalpy changes are endothermic 14, 28, and 19 J/mol. Thus, it can be concluded that the higher temperature is favourable for all protonation reactions of the ligand in the aqueous solutions, which is important for the leaching process at higher temperatures. 

As it is well known, CA is an oxygen donor, and its protonation processes are hard–hard interactions according to hard and soft acids as acceptors and bases as donors. As a rule, such a reaction is entropy-driven. The enthalpy change is positive and counteracts the protonation process. For iron, the citric complexes are yellow, light blue for Cu, light green for Ni, light violet for Co, and colourless for most REEs. Typically, three COOH groups and the OH (alcoholic) group of CA are deprotonated (and neutralized) before and during Fe, Cu, Co, and Cu complex formation. These reactions are also pH dependent. Therefore, the conditional constant gives the relationship between the concentration of formed complexes and unreacted REE ions and the complexing agent [30]. The appropriate constant can be found in [33] compared with sodium tripolyphosphate Na_5_P_3_O_10_ (STPP), ethylenediaminetetraacetic acid C_10_H_16_N_2_O_8_ (EDTA), and ethylenediaminetetra(methylenephosphonic acid C_6_H_20_N_2_O_12_P_4_ (EDTMP). They all increase with the increasing temperature. The corresponding enthalpy changes are endothermic. The entropy changes are positive in all cases, which is in agreement with the reactions between the CA and hard acceptors (such as REEs) or the borderline acceptors (such as Cu or Ni). 

Moreover, in connection with the biotoxicity of Cu, Cr, Ni, Cd, or Pb, their citrates are intensively studied. Similarly, rare earth citrates, e.g., LaCr(Cit)_2_·2H_2_O, are used as precursors in the low-temperature preparation of useful perovskite oxides. Nickel iron hexahydrate Ni_3_Fe_6_O_4_(Cit)_8_·6H_2_O is the precursor for the synthesis of ultrafine NiFe_2_O_4_ ferrites. The thermal studies of citrates of REEs for REECit·xH_2_O and REE_2_(HCit)_3_·2H_2_O types (REE = La, Ce, Pr, Nd, Sm, and Eu) were also conducted in [34,35,36,37]. The solubility of citrates of La, Ce, Pr, Nd, Eu, Gd, Tb, Dy, Ho, Er, and Y in 0.1 M (H, Na)ClO_4_ solutions was reported at 25 °C by the Skornik et al. [22]. For LaCit·3H_2_O, it was 1.96 ± 0.08 × 10^−11^, CeCit·3.5H_2_O 1.56 ± 0.05 × 10^−11^, PrCit·3.5H_2_O 1.06 ± 0.12 × 10^−11^, NdCit·3.5H_2_O 1.30 ± 0.07 × 10^−11^, EuCit·4H_2_O 0.97 ± 0.60 × 10^−11^, GdCit·4H_2_O 1.32 ± 0.04 × 10^−11^, TbCit·5H_2_O 1.51 ± 0.12 × 10^−12^, DyCit·4H_2_O 3.20 ± 0.04 × 10^−12^, HoCit·4H_2_O 2.99 ± 0. 21 × 10^−12^, and YCit·5H_2_O 0.94 ± 0.04 × 10^−11^.

As the second leaching agent, the metartaric acid C_4_H_6_O_6_ was tasted (Figure 3a,b). It is a natural organic acid found mainly in plants, which is particularly rich in grapes. It is widely applied in the food, pharmaceutical, and fracturing industries [38,39]. In [40], it was proved that TA could be used for the recovery of Mn, Li, Co, and Ni from spent lithium-ion batteries (LIBs). The third one was EDDS. EDDS (H_4_edds) is a structural isomer of EDTA (Figure 3c). It forms four isomers: *S,S*- (25%), *R,R*- (25%), *R,S*- (50%), and *S,R*- (50%). Based on the naturally occurring amino acid, i.e., L-aspartic acid, the *S,S*-isomer of EDDS is readily biodegradable. According to the OECD, 83% of S,S-EDDS converts to CO_2_ within 20 days [41]. However, the others are partly or completely non-biodegradable [42]. EDDS is also characterized by low toxicity. Therefore, all three can be called green leaching and chelating agents and can be proposed for the metal ion recovery procedure. 

As follows from our previous studies, H_2_SO_4_ as a leaching agent for the REE extraction and obtaining the PLS was quite effective (see Figure 2 and Figure 3 where the green dash line represents the values of the Fe(III) concentration, and the purple one represents Ni(II) after H_2_SO_4_ leaching under the same conditions) [30]. Analysing the leaching solution, it was found that extraction rates between 75% and 100% were obtained for heavy REE and above 100% for heavy metal ions (i.e., Fe, Co, Ni, Cu, etc.). 

As follows from the data presented in Figure 2, the determining factor in the leaching process is time. After about one hour, the leaching of individual components was observed, however, at a definitely unsatisfactory level, i.e., 24.49, 11.15, 0.01, 3.00, 305.59, 36.21, 64.12, 60.43, and 6.76 mg/L for La(III), Ce(III), Pr(III), Nd(III), Fe(III), Co(II), Ni(II), Cu(II), and Zn(II) in PLS, respectively, and increased slightly after 2 h (Figure 2a). It was only after the period of 24 h and especially after the additional grinding of samples of the black battery mass in the ball mill that the increase in leaching reached 130.22 for CA, and 51.90, 2.96, 18.60, 992.13, 126.32, 855.80, 197.19, and 32.96 mg/L for La(III), Ce(III), Pr(III), Nd(III), Fe(III), Co(II), Ni(II), Cu(II) and Zn(II), respectively (Figure 2b). Only for Fe(III) and Ni(II) was the level of leaching as satisfactory as with H_2_SO_4_.

However, the temperature effect is not crucial in this process. At 293 K, the leaching efficiency reached 105.15, 44.57, 1.93, 14.63, 1017.83, 106.18, 633.46, 184.92, and 28.91 mg/L for La(III), Ce(III), Pr(III), Nd(III), Fe(III), Co(II), Ni(II), Cu(II), and Zn(II) in PLS, respectively. At 333 K, the increase in the concentration of the individual ions was statistically insignificant and increased only by 0.01–2.2%. An increase in the initial temperature of the leaching process from 293 K to 333 K increased the leaching of metal ions from the black battery mass by 1.57% for La(III), 1.32% for Ce(III), 2.02% for Pr(III), 1.89% for Nd(III), 0.01% for Fe(III), 0.05% for Co(II), 0.09% for Ni(II), 0.55% for Cu(II), and 0.37% Zn(II), while a further increase to 343 K had no significant effect on the efficiency of the process and this temperature was not considered in father studies. This is due to the fact that the change in the protonation constants with the temperature increase is also negligible. It was proved that lower acidity at a lower temperature in the range of 288–308 K could be attributed to the formation of hydrogen bonding between the carboxylate group and the hydroxyl group, which could stabilize the deprotonated form.

Considering the addition of the complexing agent to the CA-TA and EDDS (with the possibility of mutual competition in the stage of complex formation according to the values of constants of permanently EDDS > CA > TA), higher rates of metal ion binding and thus leaching were observed at the TA addition compared to EDDS or even CA (Figure 3a,c). However, the best results were obtained for 0.3 M TA. Comparing even the same concentrations of TA and EDDS, more favourable results were obtained for TA for both 0.1 M and 0.2 M concentrations. It was found that for the analysed metal ions, the obtained concentrations were 137.89, 57.48, 3.01, 19.41, 1024.05, 122.34, 713.78, 193.78, and 27.82 mg/L for La(III), Ce(III), Pr(III), Nd(III), Fe(III), Co(II), Ni(II), Cu(II), and Zn(II) in PLS, respectively. The extraction of complexes is governed by two main factors: the competition between TA or EDDS and CA ions for the metal and the interaction of the resultant complex. It is worth mentioning that metal complexes with EDDS, TA, and CA are more stable than with sulphate ions.

It was proved that the addition of H_2_O_2_ as a leaching agent to the TA-CA and CA solutions (Figure 3b,d) did not improve the efficiency of the process. However, for the mixture of CA-TA and H_2_O_2,_ the concentration of 0.6–0.3 M and 2% H_2_O_2_ could be suggested as optimal.

It is evident that the coordination chemistry of REEs, very important in their recovery and separation, is quite different from that of *d*-transition metal ions [43]. Therefore, in the next step, the ATR-FTIR spectra of the formed complexes in the PLS with CA, TA, EDDS, and their mixtures (also in the presence of H_2_O_2_) were recorded. 

The inherent strong oxyphilicity of lanthanides causes the interaction between the binding sites -COOH, -OH (phenolic), -OH (hydroxylic), =O (carbonyl), -N (amino, imido, imino), and -S (sulfur) [44,45]. It was proved that the formation of the appropriate complexes occurs by -OH (band at 3225 cm^−1^) and -COOH (1634, 1580, and 1387 cm^−1^) both for REEs and HMs (Figure 5 and Figure 6). During the bond formation, the *4f* orbitals of Ln^3+^ ions are not active due to good shielding by the 5s^2^ and 5p^6^ orbitals. Therefore, the spectroscopic and magnetic properties depend on the ion’s properties not being affected by the field of ligands. It was also proved that they prefer anionic ligands with donor atoms of high electronegativity [43,46]. 

### 2.3. Separation Steps—Summary

After leaching, different techniques should be used for the pregnant leach solution’s (PLS) further separation [47]. Solvent extraction is an effective method for the separation of valuable elements and impurities. It involves contacting the PLS with an active extractant and appropriate modifiers dissolved in a solvent (diluent). They form immiscible phases and must have small mutual solubilities. The dissolved metal or metal complex ions and the extractant molecules undergo chemical exchange reactions, which involve cation exchange, anion exchange, or the neutral incorporation of solutes into the extractant molecules. P204, P507, and Cyanex 272 are commonly used in the extraction of Ni and Co [48]. 

P204 is C_16_H_35_O_4_P, P507 is (C_8_H_17_)_2_HPO_3_ known as 2-ethylhexyl phosphonic acid mono-2-ethylhexyl ester [49,50]. However, it was found that the aqueous acidity in the extraction and stripping for P507 was lower than that for P204 in the extraction of middle and heavy REEs [51]. Cyanex 272 has one more alkyl, one less alkoxy than P507, and its pK_a_ value is much higher than that of P507. However, due to the high cost, it is found only in a limited application in the industry for the separation of non-ferrous metals [52]. It was also proved that the Cyanex 272-P507 impregnated resin could be used for the recovery of Tm-Yb-Lu, and chloride ions do not participate in coordination. The solvent extraction of Er(III) with P507 and saponified P507 in the HNO_3_ solution and its separation from Ca(II) and Fe(III) was presented in [53]. 

Based on [54,55], 2-ethylhexylphosphonic acid mono-2-ethylhexyl ester (HEHEHP, EHEHPA, PC88A, P507, Ionquest 801) and di(2-ethylhexyl)phosphoric acid (HDEHP, D2EHPA, P204) were chosen as extractants in the industrial solvent extraction processes. EHEHPA can be proposed as an alternative to D2EHPA due to its high selectivity and lower acidity. Now P507, P507-ROH, and naphthenic acid (NA) are widely applied in China, leading to the industrial production of all individual REEs with up to 99.99% or higher purity [56,57]. For example, Sato et al. [58] reported solvent extraction using the HCl–DEHPA system. In general, the selectivity order for extracting REEs in the P204, P507, Cyanex 272-HClO_4_, HCl, HNO_3,_ or H_2_SO_4_ systems was Lu > Yb > Tm > Er > Y > Ho > Dy > Tb > Gd > Eu > Sm > Pm > Nd > Pr > Ce > La with the lgD increasing linearly with the atomic number Z. Up to now, an extraction system generally superior to P507 has not been found. However, the extracting and stripping acidities of P507 remain large, and it is still difficult to strip the HREEs, meaning that it is difficult to make high-purity heavy rare earth.

Nowadays, for the ion exchange separation of REEs, the impregnating resins and the chelating resins are mainly used. However, from a historical point of view, special attention was paid to the separation and removal of REE-nitrate complexes by means of frontal analysis from the polar organic solvent-H_2_O-HNO_3_. The affinity series of REE-nitrate complexes depends on the kind of functional groups, the kind of skeleton, the porosity of the skeleton, the cross-linking degree of the anion exchanger skeleton, as well as the kind and concentration of the polar organic solvent, the concentration of nitric acid, the addition of another organic solvent, and the concentration of rare earth(III) elements. Since the sixties of the last century, mainly cation exchangers and the elution process by complexing agents have been used for the separation of REEs using ion exchange methods. In this process, the order of elution for individual REEs depends on the values of stability constants of formed complexes. They generally increase from light rare earth elements (LREEs) to heavy rare earth elements (HREEs). The ion exchange of REEs in the presence of chelating ligands on the anion exchangers is still a poorly studied area. As for the isotopes of these elements, the separation processes were mainly of an analytical or physicochemical character. 

## 3. Materials and Methods

### 3.1. Materials

The leaching agent’s citric acid C_6_H_8_O_7_ (CA), C_4_H_6_O_6_ metatartaric acid (TA), and ethylenediaminedisuccinic acid (EDDS) were used. 

The non-sterile citric acid (CA, 0.6 M, Helmholtz-Centre for Environmental Research-UFZ, Leipzig, Germany) production process was performed in a fed-batch mode in a stirred tank reactor (ISF215, Infors AG, Bottmingen, Switzerland). The bioreactor was inoculated with a 10% (v/v) pre-culture of *Y. lipolytica H181* using the waste frying oil as the substrate and was obtained by the Helmholtz-Centre for Environmental Research-UFZ (Germany). The optimized process conditions for the CA production were as follows: pH 5.0, temperature 30 °C, dissolved oxygen concentration > 20%, stirring speed 800 rpm, and airflow of 4 L/min. After the separation of yeast biomass and waste frying oil residues were measured, the following concentrations in the permeate and the sustainable form of CA solution were used for the whole experimental work on the sorption of REE: citric acid (CA) 186.74 g/L (2R,3S)-isocitric acid 8.5 g/L, Cl^−^ 0.36 g/L, Ca^2+^ 0.63 g/L, K^+^ 0.08 g/L, Na^+^ 23.19 g/L, Mg^2+^ 0.03 g/L, SO_4_^2−^ 2.04 g/L. The determination of organic acids, inorganic anions, and cations was made by the ion chromatography (IC) system ICS 5000+ DP (Thermo Fisher Scientific GmbH, Dreieich, Germany) [28,29,30]. Metatartaric acid (CAS No. 56959-20-7, ≥90%) was supplied by Begerow (Poland), and the trisodium salt of ethylenediaminedisuccinic acid was produced on the commercial scale by the firm Innospec Inc. (CAS No. 20846-91-7, ≥90%). The natural route involves the recovery of tartaric acid from wine. The synthetic route of metatartaric acid manufacturing involves the chemical reactions of maleic anhydride. Its derivatives can also be obtained microbially from cis-epoxysuccinic acid using various microorganisms. S,S-EDDS is a biodegradable complexing agent that is a structural isomer of EDTA (ethylenediaminetetraaceitic acid). All substances were of the greatest analytical quality.

### 3.2. Leaching Process

The materials tested included spent R6 (AA standard) nickel metal hydride (NiMH) batteries from the selective collection points located in the Mazowieckie Voivodship (Poland). The spent crushed NiMH batteries were provided by the industrial battery recycling operator. Their different components were separated, classified, and weighed. The active materials from the cathode and anode were manually removed and leached. The obtained fractions are presented in Figure 7.

After separation into 1 g fractions of raw materials in the form of a black mass, it was leached in 100 mL glass flasks using the mechanical shaker type 358 A (Elpin+, Poland) with a constant vibration amplitude (8 units) for 1, 2, and 4 h at different temperatures of 20, 40, and 60 °C and at the constant solid to liquid (S/L) ratio of 1:10. For the removal of REE, the following acids (also with the H_2_O_2_ addition) were used: citric acid (CA) and CA with metatartaric (TA) or ethylenediaminedisuccinic acid (EDDS). After the vacuum filtration, the REE-rich solutions were separated. The filtrate pH was determined using the pHmeter pHM82 (Radiometer, Denmark). The samples of the black battery mass remains were washed several times with distilled water. The obtained filtrates were transferred quantitatively to the volumetric flasks and diluted, and the obtained solutions were analysed concerning the Fe(III), Co(II), Ni(II), Cu(II), and Zn(II) content and using the atomic absorption spectrometer (AAS, Varian AAS 240 FS) at 259.240 nm, 238.892 nm, 216.555 nm, 327.395 nm, and 206.200 nm, respectively. The solution was found to be rich in Fe, Ni, Co, Cu, and Zn as the main impurities. Moreover, trace amounts of Cd were found. The concentrations of REEs and heavy metal ions were determined by the inductively coupled plasma optical emission spectroscopy ICP-OES method using the Varian 720-ES axial ICP-OES (Varian Inc., USA). The ICP-OES operating conditions were: a power of 1.0 kW, an optical resolution of 0.004 nm, a plasma gas flow of 15.0 L/min, a pump speed of 15 rpm, a replicate read time of 10 s, a sample uptake delay of 18 s and 3 replicates. La(III), Ce(III), Pr(III), and Nd(III) ions were determined at the wavelengths of 333.749 nm, 446.021 nm, 410.072 nm, and 401.224 nm, respectively. Furthermore, the effect of pH on leaching was studied by adjusting the pH. The pH was adjusted using nitric acid or sodium hydroxide. The control samples were prepared for a solid-to-liquid (S/L) ratio of 1:10 for the spent NiMH battery black mass powder and ultrapure water without any leaching agents. The investigations were carried out in three replicates, and the errors were recorded. The maximum error was found to be ±7%. All analyses were made using ORIGIN PRO 8SRO software (v.8.0725).

After leaching the FT-IR spectra, the sample was recorded with the Agilent Cary 630 FT-IR spectrometer (Agilent Technologies, Inc., Santa Clara, CA, USA) using the ATR technique in the wavelength range of 500–4000 cm^−1^ with the spectral resolution of 4 cm^−1^. The sample spectra were recorded using the Micro Lab FTIR software (version B.04). Additionally, the Agilent Resolutions Pro software (version 5.2.0.861) was used for the post-collect analysis.

The X-ray fluorescence analysis (XRF) was performed using the Canberra Packard to identify the crystalline phases in the battery black mass. The characteristics of the spectrometer are the excitation of the X-ray fluorescence of the sample atoms using the radioisotopic sources: Cd-109, Fe-55, Am-241, the energy dispersive analysis method providing high sensitivity measurements and cumulative spectral recording for various elements, the Si (Li) semiconductor detector operating at a liquid nitrogen temperature, the multi-channel X-ray fluorescence photon energy analyzer, as well as the computer control and the data processing using the Micro AXIL and QXAS software (System 100 V3.0).

## 4. Conclusions

Battery recycling is a crucial tool for lowering carbon emissions and achieving the circular economy-focused regulatory standards anticipated by the proposed EU Battery Regulation. The performance of ‘green’ complexing agents such as citric (CA), metatartaric (TA), and ethylenediaminedisuccinic acid (EDDS) (also with H_2_O_2_ addition) was investigated in the recovery or REEs and HMs from PLS. The influence of a phase contact time (1, 2, 4 and 240 h), solution pH (4.45–4.47), temperature (393–333 K), as well as CA, TA, and EDDS concentration (0.6 M and 0.1–0.3 M), was studied. In our studies, it was found that the period of 24 h, especially after the additional grinding of samples using 0.6 M CA, is suitable for leaching with 130.22, 51.90, 2.96, 18.60, 992.13, 126.32, 855.80, 197.19, and 32.96 mg/L of La(III), Ce(III), Pr(III), Nd(III), Fe(III), Co(II), Ni(II), Cu(II), and Zn(II), respectively. However, the best results were obtained using 0.3 M TA. Comparing even the same concentrations of TA and EDDS addition to CA, more favourable results were obtained for TA for both 0.1 M and 0.2 M concentrations. It was found that for the analysed metal ions, the obtained values were equal to 137.89, 57.48, 3.01, 19.41, 1024.05, 122.34, 713.78, 193.78, and 27.82 mg/L for La(III), Ce(III), Pr(III), Nd(III), Fe(III), Co(II), Ni(II), Cu(II), and Zn(II), respectively. The leaching process of the black battery mass allowed for a near-quantitative 100% recovery of HMs ions and more than 75% recovery of REEs ions. This efficiency was achieved using 0.6 M CA. An equally high leaching rate was obtained in the 0.6 M CA-0.3 M TA system at pH 4.47 over 24 h. The proposed set of parameters should be viewed as a base tool that can be developed, modified, or tailored to obtain the REE-based PLS and then a specific way of REEs recovery. They provide a much smaller environmental impact compared to the other REE recycling methods.

## Figures and Tables

**Figure 1 molecules-28-00965-f001:**
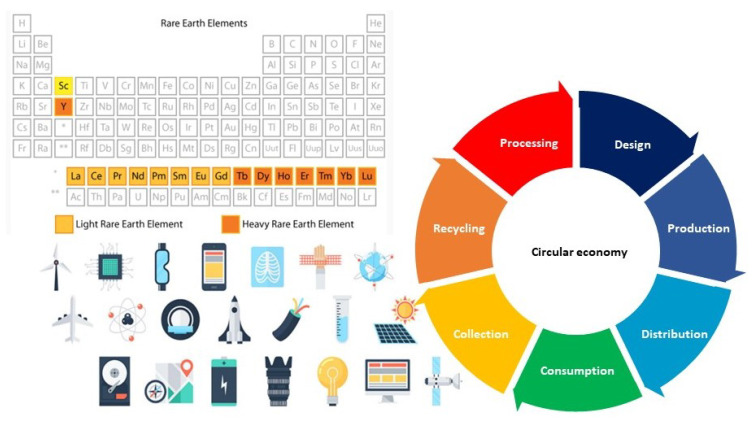
Circular economy and REEs recovery.

**Figure 2 molecules-28-00965-f002:**
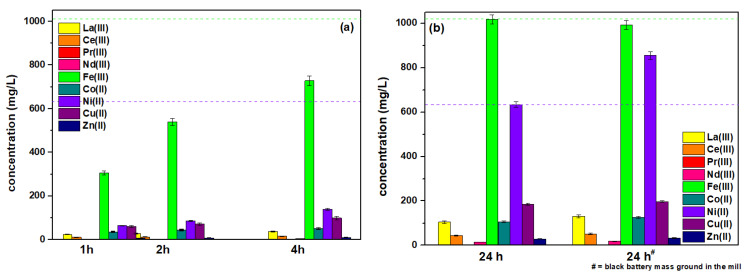
Concentration of La(III), Ce(III), Pr(III), Nd(III), Fe(III), Co(II), Ni(II), Cu(II), and Zn(II) in the PLS obtained from the black battery mass using 0.6 M CA (**a**) Without and (**b**) After the additional grinding (#) after 1, 2, 4, and 24 h leaching. The green dash line represents the values of Fe(III) concentration and the purple one represents Ni(II) after H_2_SO_4_ leaching under the same conditions.

**Figure 3 molecules-28-00965-f003:**
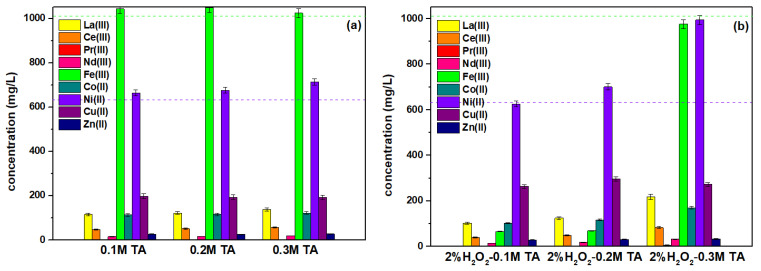
The concentration of La(III), Ce(III), Pr(III), Nd(III), Fe(III), Co(II), Ni(II), Cu(II), and Zn(II) in the PLS obtained from the black battery mass using 0.6 M CA with the additional amount of (**a**) TA, (**b**) H_2_O_2_ and TA, (**c**) EDDS as well as (**d**) H_2_O_2_ at 24 h. The green dash line represents the values of Fe(III) concentration and the purple one represents Ni(II) after H_2_SO_4_ leaching under the same conditions.

**Figure 4 molecules-28-00965-f004:**
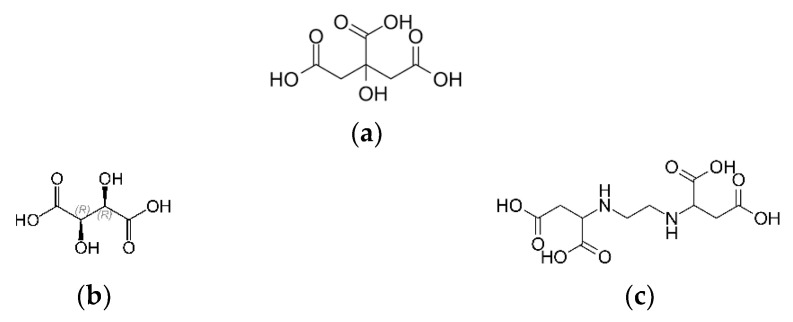
The (**a**) Citric, (**b**) Metatartaric, and (**c**) Ethylenediaminedisuccinic acids.

**Figure 5 molecules-28-00965-f005:**
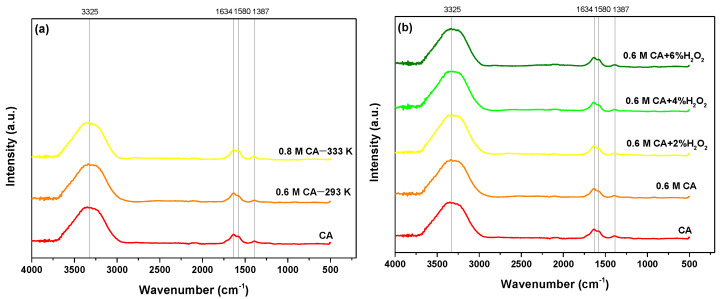
FTIR spectra of PLS solutions obtained using (**a**) 0.6 M CA and (**b**) With the additional amount of H_2_O_2_.

**Figure 6 molecules-28-00965-f006:**
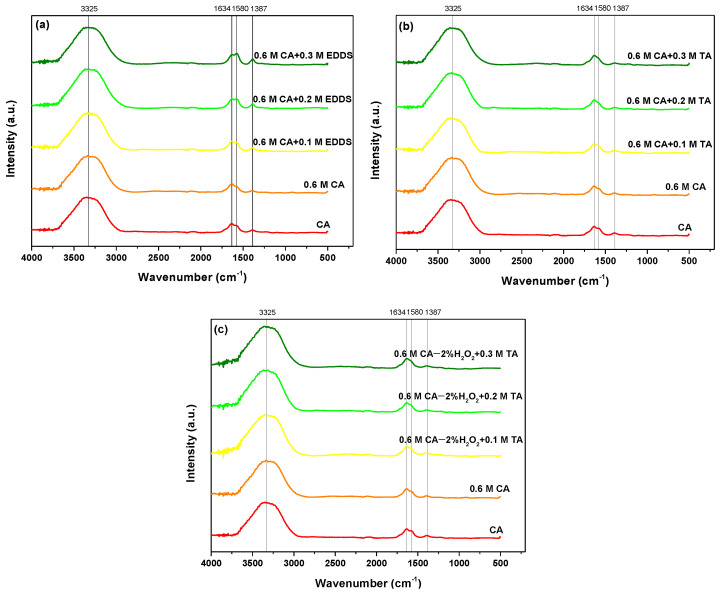
FTIR spectra of PLS solutions obtained using (**a**) 0.6 M CA with EDDS, (**b**) 0.6 M CA with TA, and (**c**) 0.6 M CA with TA with the additional amount of H_2_O_2_.

**Figure 7 molecules-28-00965-f007:**
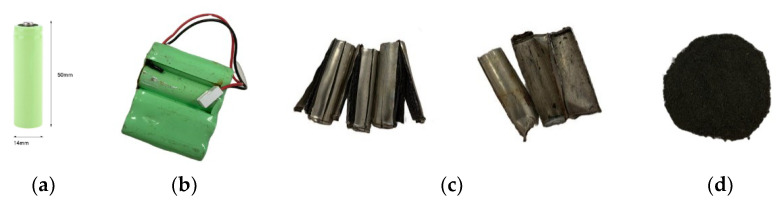
Spent NiMH batteries (**a**) The size of the typical NiMH battery, (**b**) Cases of spent cells, (**c**) Housings and small plastic parts, (**d**) Black battery mass.

**Table 1 molecules-28-00965-t001:** Chemical composition of battery black mass (%).

Composition	Fe	Ni	Co	Cu	Zn	La	Ce	Pr	Nd
Content (%)	11.293	30.769	3.414	3.268	0.787	4.306	1.839	0.128	0.551
**Composition**	**O**	**Al**	**Si**	**K**	**Mn**	**Ca**	**Cl**	**Mg**	**Y**
Content (%)	26.015	5.035	2.602	1.589	1.046	1.036	0.553	0.190	0.122

**Table 2 molecules-28-00965-t002:** Parameters of black battery mass leaching using CA, TA, and EDDS.

Parameter	CA	CA-TA	CA-EDDS
**leaching solution (M)**	0.6	0.1, 0.2, 0.3	0.1, 0.2, 0.3
**with H_2_O_2_ addition**	2, 4, 6%	2%	-
**pH**	4.45–4.47	3.72–4.21	4.81–5.50
**time (h)** #with black battery mass ground in the mill	1, 2, 4, 24, 24#	1, 2, 4, 24, 24#	1, 2, 4, 24, 24#
**temperature (K)** **Pregnant leach solution (PLS)**	293 and 333	293	293
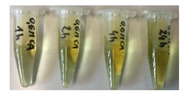	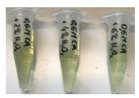		

## Data Availability

Data are contained within the article.

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
