# Peer review of "Green Extractants in Assisting Recovery of REEs: A Case Study"

_molecules, 2023, doi:10.3390/molecules28030965_

Round 1
Reviewer 1 Report
1. Abstract: (a) In line 17, it may be indicated that study was done on what?, (b) It is simply repetition of lines 249-258, and (c) Entire abstract should be made informative.
2. Objectives should be clearly highlighted under section 'Introduction'.
3. Instead of 1.1, an independent Section should be given for 'Recycling Market'.
4. Repetition can be avoided by proper editing of manuscript.
Author Response
1.Abstract: (a) In line 17, it may be indicated that study was done on what?, (b) It is simply repetition of lines 249-258, and (c) Entire abstract should be made informative.
Reply: The Abstract was changed according to the suggestions. All changes were marked red.
- Objectives should be clearly highlighted under section 'Introduction'.
Reply: The aims of the paper were changed.
Modification: Therefore, in the paper the research on the applicability of different types of com-plexing agents for leaching of REEs from spent NiMH batteries is carried out. The main aim is determination of the influence of type of leaching system containing citric (CA), metatartaric (TA), ethylenediaminedisuccinic acid (EDDS) (also with the H2O2 addi-tion) on effectiveness of REEs leaching. The studies are conduct considering the effects of the phase contact time, initial concentration of CA, TA and EDDS as well as H2O2, pH and temperature. After the comparisons the most suitable treatment way was se-lected. To achieve the essential aim, semi-technical scale experiments are being planned.
- Instead of 1.1, an independent Section should be given for 'Recycling Market'.
Reply: The section of 1.1. was removed.
- Repetition can be avoided by proper editing of manuscript.
Reply: The manuscript was checked and modified. All repetitions were removed.
Reviewer 2 Report
The introduction should have been more concise.
I did not see 1.2 at the end of section 1.1, which I think is inappropriate, and there are similar problems with 3.1.
How to deal with the error in the experiment should be explained in the article.
Some graphs and tables should be resized appropriately.
The author's narration in the results and discussion section should be more organized.
Author Response
- The introduction should have been more concise.
Reply: The introduction was changed. All repetition were removed.
- I did not see 1.2 at the end of section 1.1, which I think is inappropriate, and there are similar problems with 3.1.
Reply: The numbering of paragraphs has been changed.
- How to deal with the error in the experiment should be explained in the article.
Reply: The description in the part 2.2. Leaching process was added.
- Some graphs and tables should be resized appropriately.
Reply: The graphs and tables were prepared in accordance with the guidelines.
- The author's narration in the results and discussion section should be more organized.
Reply: The last part of the paper was changed – please see all corrections marked red.
Reviewer 3 Report
1. Introduction is too long, please revise again and focus on the extraction of REEs from NiMH.
2. The concentration of REE in the NiMH sample is low, which is not reliable to estimated only by XRF. It is better to be determined by ICP, and the concentration in sample and the leach liquor can be used for evaluate the leaching efficient (%). Therefore, it should show the results in term of leaching efficiency, not only by the concentration of REEs.
3. There are 2 Figure 5. Please revise in text as well.
4. The authors mentioned the leaching of REE depends on the complexity of systems: type of REEs, type of leachant, pH and temperature. However, the results and interpretations are not clear and efficient to prove this statement. For example, it should show the different leaching in different acids, pH, and temperature, and explain if possible, and conclude the optimal conditions.
5. Line 448-553: the authors describe the separation of REEs; however, it is too long. It is better to discuss more the advantages and disadvantages of present work since the investigation is the using of green extraction to dissolve REEs.
6. Please revise the English through the manuscript.
Author Response
- Introduction is too long, please revise again and focus on the extraction of REEs from NiMH.
Reply: The part Introduction was rearranged. All changes were marked red.
- The concentration of REE in the NiMH sample is low, which is not reliable to estimated only by XRF. It is better to be determined by ICP, and the concentration in sample and the leach liquor can be used for evaluate the leaching efficient (%). Therefore, it should show the results in term of leaching efficiency, not only by the concentration of REEs.
Reply: The concentration of all metal ions was determined by ICP-OES method. The concentrations of REEs and heavy metal ions were determined by the inductively coupled plasma optical emission spectroscopy ICP-OES method using the Varian 720-ES axial ICP-OES (Varian Inc., USA). The ICP-OES operating conditions were: the power 1.0 kW, the optical resolution 0.004 nm, the plasma gas flow 15.0 L/min, the pump speed 15 rpm, the replicate read time 10 s, the sample uptake delay 18 s and replicates 3. La(III), Ce(III), Pr(III), and Nd(III) ions were determined at the wavelengths of 333.749 nm, 446.021 nm, 410.072 nm, and 401.224 nm, respectively. The obtained result were presented in Figs. 3 and 4. However, as for black battery mass it is impossible to fully dissolve it to establish the metal ions contents. Initially, waste batteries are collected, sorted, discharged and disassembled. This is followed by mechanical crushing, drying, sorting sieving and pyrolysis to 700°C to remove any remaining electrolyte and potentially hazardous to health fluorine-containing components.
The resulting material is what is referred to in the battery recycling industry as ‘black mass’. The black mass contains large quantities of the main metals used in the production of cathode materials: lithium, nickel, cobalt, and manganese. These critical materials can then be extracted from the black mass and reused in new battery production or new products and/or applications. The black mass can be the feedstock for the commercial hydrometallurgical refineries for battery recycling. It can be analyzed using:
- X-ray fluorescence analysis (XRF)
- a visual examination and binocular microscopy, manual scanning electron microscopy (SEM), automated scanning electron microscopy with linked energy dispersive spectrometers (SEM-EDS),
- advanced mineral identification and characterization systems (AMICS), X-ray computed tomography (X-CT)
- or laser ablation inductively-coupled plasma mass spectrometry (LA-ICP-MS).
In our case the first one was chosen.
- There are 2 Figure 5. Please revise in text as well.
Reply: The numeration of Figure 5 was changed. Other figures were re-numerated.
- The authors mentioned the leaching of REE depends on the complexity of systems: type of REEs, type of leachant, pH and temperature. However, the results and interpretations are not clear and efficient to prove this statement. For example, it should show the different leaching in different acids, pH, and temperature, and explain if possible, and conclude the optimal conditions.
Reply: The examples of the differences of the CA, TA and EDDS are presented in Figs.3a-b and Figs.4a-d. The comments were presented in the text. The discussion of the obtained results was improved.
- Line 448-553: the authors describe the separation of REEs; however, it is too long. It is better to discuss more the advantages and disadvantages of present work since the investigation is the using of green extraction to dissolve REEs.
Reply: The mentioned part was rearranged.
- Please revise the English through the manuscript.
Reply: The paper was checked by the native speaker.
Reviewer 4 Report
The authors have written an article entitled “Green extractants in assisting recovery of REEs: A Case study”. The manuscript is quite interesting, well framed, and based on the application of ‘green’ extractants such as citric (CA), metatartaric (TA), ethylenedia- minedisuccinic acid (EDDS) (also with H2O2 addition) for recovery of REEs was studied. The work reported in this manuscript is interesting and well-presented. The article has some grammatical and sentence errors, and the language organization needs to be improved. The authors have described the concept to a greater extent but the manuscript still needs some Minor corrections before publishing in the Molecules.
I advise the authors to consider the following points when revising their manuscript.
Comment 1: Abstract should clearly discuss the problem statement, so it should be revised.
Comment 2: The manuscript must be checked for typographical/ grammatical, superscript, and subscript errors.
Comment 3: Include the all chemicals/materials details such as purity, manufacturer origin, etc.,
Comment 4: Improve the Figure 1 Resolution.
Comment 5: Overall, the “Discussion” section needs more comparative study than just mentioning the obtained results. So please compare and discuss the obtained results with previous studies.
Comment 6: Please improve the conclusion with clear quantitative findings and more emphasis on the findings and its implication may be mentioned in the conclusion section.
Comment 7: The homogeneity of the reference section needs to be maintained. So please check and revise accordingly to the journal's instructions.
Author Response
- Abstract should clearly discuss the problem statement, so it should be revised.
Reply: The Abstract was changed according to the suggestions.
- The manuscript must be checked for typographical/grammatical, superscript, and subscript errors.
Reply: The paper was checked by the native speaker. All changes were marked red.
- Include the all chemicals/materials details such as purity, manufacturer origin, etc.,
Reply: The paragraph 2.1. Materials was modified and useful data were added.
- Improve the Figure 1 Resolution.
Reply: The resolution was changed.
- Overall, the “Discussion” section needs more comparative study than just mentioning the obtained results. So please compare and discuss the obtained results with previous studies.
Reply: The results were compared with the results with previous studies.
6: Please improve the conclusion with clear quantitative findings and more emphasis on the findings and its implication may be mentioned in the conclusion section.
Reply: The conclusion part was changed.
- The homogeneity of the reference section needs to be maintained. So please check and revise accordingly to the journal's instructions.
Reply: References were checked and revised.
Round 2
Reviewer 1 Report
In line 310, 11,29% of iron and 30,77% of nickel should be corrected to 11.29% iron and 30.77% nickel.
Reviewer 2 Report
The manuscript can be acceptable after the revision.
Reviewer 3 Report
Please revise the heading title in session 3.